# Systolic Blood Pressure and Pulse Pressure Are Predictors of Future Cardiovascular Events in Patients with True Resistant Hypertension

**DOI:** 10.3390/diagnostics13101817

**Published:** 2023-05-22

**Authors:** J. Mesquita Bastos, Lisa Ferraz, Flávio G. Pereira, Susana Lopes

**Affiliations:** 1School of Health Sciences and Institute of Biomedicine-iBiMED, University of Aveiro, 3810-193 Aveiro, Portugal; 2Cardiology Department, Centro Hospitalar do Baixo Vouga, 3810-164 Aveiro, Portugal; 3Internal Medicine Department, Centro Hospitalar do Baixo Vouga, 3810-164 Aveiro, Portugal; 4Polytechnic of Coimbra, ESTeSCoimbra Health School, Physiotherapy Department, 3040-854 Coimbra, Portugal

**Keywords:** resistant hypertension, ambulatory blood pressure, systolic blood pressure, pulse pressure, cardiovascular risk, cardiovascular prognosis

## Abstract

Given the increased risk of cardiovascular events associated with resistant hypertension, predictive cardiovascular prognosis is extremely important. Ambulatory blood pressure monitoring (ABPM) is mandatory for resistant hypertension diagnosis, but its use for prognosis is scarce. This observational longitudinal study included 258 patients (mean age of 60.4 ± 11.2 years; 61.2% male), who underwent 24 h ABPM in a hypertension unit from 1999 to 2019. The outcomes were global cardiovascular events (cerebrovascular, coronary, and other cardiovascular events). The mean follow-up period was 6.0 ± 5.0 years. Sixty-eight cardiovascular events (61 nonfatal) were recorded. Patients who experienced cardiovascular events were generally older, with higher rates of chronic kidney disease and prior cardiovascular events. The 24 h systolic blood pressure (hazard ratio 1.44; 95% CI 1.10–1.88), night systolic blood pressure (1.35; 95% CI 1.01–1.80), and 24 h pulse pressure (2.07; 95% CI 1.17–3.67) were independent predictors of global cardiovascular events. Multivariate Cox analysis revealed a higher risk of future cardiovascular events, particularly in patients with a 24 h daytime and nighttime pulse pressure > 60 mm Hg with respective hazard ratios of 1.95; 95% CI 1.01–3.45; 2.15; 95% CI 1.21–3.83 and 2.07; 95% CI 1.17–3.67. In conclusion, APBM is a fundamental tool not only for the diagnosis of resistant hypertension, but also for predicting future cardiovascular events.

## 1. Introduction

Cardiovascular (CV) and cerebrovascular events are mostly caused by arterial hypertension. Epidemiological studies report a 7.1 million-death incidence rate annually, with ischemic illness accounting for 49% of deaths and cerebrovascular disease for 69% of deaths globally [1]. The worst outcome is associated with resistant hypertension (RH). It is expected that the prevalence of RH will increase in the coming years due to the increase in the world population, obesity, and the prevalence of diabetes mellitus (DM) [2,3,4]. Independent of BP control, RH is associated with a higher risk of CV disease and all-cause mortality [2,3,5,6]. Patients with RH also have a higher prevalence of end organ damage [7], DM, chronic kidney disease (CKD) patients [2,3,5], and obesity [8,9,10]. The estimated prevalence of RH among treated arterial hypertension patients is approximately 10–18% [2,5,11]. However, prevalence rates range from 5–30% in patients with treated hypertension, due to the different definitions of RH used [5,12]. Resistant hypertension is defined as uncontrolled blood pressure (BP), despite the administration of optimum doses of three first-line classes of antihypertensive drugs, including a diuretic (renin-angiotensin system blockers, calcium-channel blockers, and thiazide diuretics) or adequate BP control requiring four or more antihypertensive drugs from different classes [13]. According to the European Society of Hypertension, RH is commonly diagnosed based on office BP that should be confirmed by ambulatory blood pressure monitoring (ABPM) or home blood pressure monitoring (HBPM) [14]. ABPM is key, given it removes the often-observed white coat effect, thus creating a more real and homogeneous sample [15]. In the study by Sierra et al. (2011), a group of over 68,000 treated hypertensive individuals included in the Spanish ABPM registry were analyzed. Based on office measurements, the prevalence of RH was 14.8% among treated hypertensives and 12.0% when only patients with BP ≥140/90 mmHg were included (i.e., excluding patients with normal BP but treated with ≥4 antihypertensive drugs). However, after the assessment of the ABPM data, the prevalence of RH changed dramatically. Surprisingly, 37.2% of the originally identified RH patients had ‘white-coat RH’ (24 h systolic blood pressure (SBP)/diastolic blood pressure (DBP) < 130/80 mmHg). Whereas, only 62.5% had true RH (24 h SBP/DBP ≥ 130/80 mmHg) [16]. Similar results were reported by Parati et al. (2014), who found that less than 40% of the patients fulfilled the diagnostic criteria for RH based on ABPM data, compared to office BP due to the ‘white coat effect’ [15] This difference in the use of ABPM data or office reduces the comparability of these studies’ results [15,17,18,19,20,21,22,23]. Besides the stated importance of ABPM data for the diagnosis of RH, it is also an important tool for CV prognosis in patients with hypertension [14,15]. Some studies have found predictive value in ABPM for the onset of target organ damage and future CV events. Furthermore, future CV events were associated with higher daytime ABPM values [17,24,25], higher nighttime BP levels [26,27], and more specifically, anomalous nighttime dipping patterns [24,27]. In addition, pulse pressure (PP), (the difference between SBP and DBP values) is also a well-established risk factor for CV events among the hypertensive population [28,29,30]. In middle-aged and older people, PP is increased and has additional adverse prognostic significance. Specifically, in older adults, a PP greater than 60 mmHg is indicative of CV risk [14]. However, its value for patients with a PP greater than 60 mmHg is yet to be determined. In this study, we aimed to determine which variables were the most accurate predictors of future cardiovascular events when the diagnosis of RH was based on ABPM data, excluding the “white coat effect”.

## 2. Materials and Methods

### 2.1. Study Design and Patients

This retrospective study included patients aged 18 years or older who underwent 24 h ABPM between 1999 and 2019 in the hypertensive ambulatory unit of the Centro Hospitalar do Baixo Vouga Aveiro, Portugal. All included patients had a mean 24 h SBP of ≥130 mmHg or daytime SBP ≥ 135 mmHg during the day, while taking maximally tolerated doses of at least three antihypertensive agents, including a diuretic [6], or controlled BP with four or more antihypertensive agents [31]. Secondary hypertension was an exclusion criterion. The ethics committee of the Centro Hospitalar do Baixo Vouga Aveiro approved the study (N/Ref. 073619, 21 September 2016). All procedures were conducted in accordance with the Helsinki Declaration.

### 2.2. Clinical Data

With the 24 h ABPM data, age, sex, body mass index (BMI), and a list of antihypertensive drugs in use were collected. In addition, the medical files of all patients were consulted to collect data regarding CV risk factors, such as DM, smoking history, history of previous CV events, blood glucose, dyslipidaemia, serum creatinine, presence of proteinuria, and low-density lipoproteins cholesterol (LDL) and echocardiogram. Glomerular filtration rate (GFR) was calculated according to the Cockcroft–Gault equation, and CKD was assumed if the patient had GFR <60 mL/min/1.73 m^2^, according to the 2021 kidney disease guidelines [31]

### 2.3. Events

The type and date of the events were collected from the hospital’s records.

CV events were subdivided in coronary events (myocardial infarction, coronary angioplasty, coronary by-pass, angina pectoris), sudden death, acute heart failure requiring hospitalization, cerebrovascular events (ischemic and hemorrhagic strokes, transient ischemic attack), and peripheral arterial disease. When more than one event occurred in the same patient, only the first event was considered and at that moment, for the purposes of this study, the follow-up period was considered to have ended. Then, events were classified into fatal and nonfatal. For all deaths, the cause of death was confirmed by consulting hospital records or death certificates and classified as CV or other. When the cause of death was not specified, it was recorded as undetermined. Whenever patients did not present any documented event in the medical file, the follow up ended with the last registered assessment.

### 2.4. ABPM

ABPM was performed using the Spacelabs 90,207 device. All patients underwent 24 h BP monitoring on a weekday, with measurements on the nondominant arm every 15 min during the day and every 30 min at night. Recordings with >70% valid data were accepted, with >20 valid readings while awake with at least 2 valid readings per hour and at >7 valid readings while asleep with at least 1 valid reading per hour. The patient should not exercise vigorously. At the time of inflation, the patient should stop moving and keep the arm relaxed [15]. The nocturnal SBP dipping (%) was calculated as 100 × [1 − sleep SBP/awake SBP ratio]. According to this, patients were classified as extreme dippers (SBP decline >20%), dippers (SBP decline more than 10% and less than 20%), nondippers (SBP decline between 0 and 10%), and risers (increase in SBP during nighttime) [32]. PP was defined as the difference between systolic and diastolic values [14,15,33].

### 2.5. Statistical Analysis

Statistical analyses were performed using SPSS version 25.0 (SPSS Inc., Chicago, IL, USA). Continuous variables are presented as mean ± standard deviation. Differences between groups were assessed by parametric (*t*-test) or equivalent nonparametric tests as appropriate. Differences in proportions were assessed using the chi-square test. Long-term cumulative survival curves in PP, with a cut off of 60 mmHg, were estimated using the Kaplan–Meier method, and comparisons between the two groups were made using a log-rank test. The effects of prognostic factors on survival were assessed using hazard ratios (HR) determined by univariate and multivariate regression analysis. First, ABPM variables (SBP, DBP and PP) for the 24 h period, daytime and nighttime, as well as night-to-day ratios were assessed in univariate Cox regression analysis. Those showing significant associations were then entered in a model including confounding variables (age, sex, body mass index, DM, previous CV event) for the multivariate Cox analysis.

For ABPM variables, a 1 standard deviation (SD) increment was used to report HR (95% confidence interval). Statistical significance was defined as a two-sided *p* value < 0.05.

## 3. Results

Between 1999 and December 2019, ABPM was performed in 4501 patients. Of these, 258 (5.7%) were patients with RH. Most patients were male (*n* = 158, 61.2%), with a mean age of 60.4 ± 11.2 years. The mean follow-up period was 6.0 ± 5.0 years. All patients were taking three or more antihypertensive drugs at the highest tolerated dose, with a mean number of 4.1 ± 0.8 drugs. Eighty-nine patients (34.4%) had a previous CV event, and most presented CV risk factors (dyslipidemia, 74.2%; obesity, 45.2%; DM, 45.2%; CKD, 31.3%). Table 1 presents patients characteristics.

Of the 68 CV events, 25 were cerebrovascular events (20 ischemic strokes and 5 hemorrhagic strokes), 20 coronary events, and 22 other CV events (16 acute heart failures, 6 peripheral arterial disease) and 1 sudden death (Table 2). Regarding the 18 deaths during follow up, 7 were considered CV events (2 ischemic cerebrovascular events, 3 coronary events, 1 acute heart failure, and 1 sudden death), and 11 had nonCV causes (8 unknown causes, 1 cancer, and 2 infectious causes).

Table 3 compares patients with no CV events during follow up to those who experienced CV events. Patients with CV events were older (63.2 vs. 59.3 years, *p* < 0.05) and had a lower GFR (63.1 ± 32.9 vs. 74.7 ± 33.2 mL/min/m^2^, *p* < 0.05). Patients with CV events also had a significantly higher prevalence of previous cardiovascular events, (52.9% vs. 27.7%, *p* < 0.05) and CKD (52.5% vs. 31.2%, *p* < 0.05), compared to patients with no CV events.

The 24 h SBP (138.9 ± 16.4 vs. 132.8 ± 16.1 mmHg, *p* < 0.05), daytime SBP (143.3 ± 16.1 vs. 137.7 ± 17 mmHg, *p* < 0.05), and nighttime SBP (130.1 ± 17.6 vs. 123.7 ± 17.9 mmHg, *p* < 0.05), were higher in patients with events. Likewise, 24 h PP (62.3 ± 14.7 vs. 55.4 ± 12.9 mmHg, *p* < 0.05), daytime PP (63.0 ± 14.9 vs. 56.2 ± 13.6 mmHg, *p* < 0.05), and nighttime PP (60.8 ± 13.8 vs. 53.8 ± 13.0 mmHg) were also higher in patients who experienced a CV event, compared to patients with no events.

Figure 1 presents the Kaplan–Meier survival curve free of events for the cut off of ambulatory PP > 60 mmHg. Patients with a 24 h PP > 60 mmHg (log rank 12.1; *p* < 0.001), daytime PP > 60 mmHg (log rank 13.5; *p* < 0.001), and nighttime PP > 60 mmHg (log rank 19.3; *p* < 0.001) presented lower survival.

For the univariate Cox analysis, the confounding factors age, sex, BMI, diabetes, previous CV events, the number of antihypertensive medications, the ejection fraction of the left ventricle, LDL, glycemia, and creatinine were evaluated. Only age, gender, BMI, diabetes mellitus, and previous cardiovascular events were statistically significant and considered for the univariate and multivariate Cox analyses (Table 4).

In the univariate Cox analysis, the 24 h SBP (HR 1.37, CI 1.08–1.72) and night SBP (HR 1.45, CI 1.13–1.88) were predictors of CV events. All PP variables, namely 24 h PP (HR 1.51, CI 1.20–1.59), daytime PP (HR 1.44, CI 1.13–1.82), and nighttime PP (HR 1.45, CI 1.13–1.80) were associated with CV events. Furthermore, 24 h PP > 60 mmHg, daytime PP > 60 mmHg, and nighttime PP > 60 mmHg were also significant predictors of CV events (Table 4).

In a multivariate Cox analysis, the 24 h SBP (HR 1.44, CI 1.10–1.88), nighttime SBP (HR 1.35, CI 1.01–1.80), and the 24 h PP (HR 1.39, CI 1.02–1.89) were independent predictors of CV events. When the PP > 60 mmHg was considered in the multivariate Cox, the 24 h PP > 60 mmHg (HR 1.95, CI 1.01–3.45), daytime PP > 60 mmHg (HR 2.15, CI 1.21–3.82), and nighttime PP > 60 mmHg (HR 2.07, CI 1.17–3.67) were significant predictors of CV events (Table 4)

## 4. Discussion

The present study was to determine which variables were the most accurate predictors of future cardiovascular events when the diagnosis of RH was based on ABPM data, excluding “white coat resistant hypertension”. The main findings are: (1) patients who experienced CV events were older, male, more likely to have experienced previous CV events, and with lower creatinine clearance; (2) the 24 h SBP, daytime SBP, and nighttime SBP are predictors of CV events (both fatal and nonfatal), but not DBP values; and (3) the cut off value of PP > 60 mmHg was associated with a higher risk of CV events. It is well documented that RH is associated with a higher CV risk (older age, male, obesity, DM, CKD) and worse outcomes [2,3,5,6,34,35,36] A retrospective study involving more than 200,000 participants observed a higher prevalence of coronary events (24%), strokes (14%), and heart failure (46%) in RH patients than in other types of patients with hypertension [6]. Smith et al. (2014) reported that patients with true RH may have a greater BP burden over time, contributing to a higher CV risk, compared to those with controlled arterial hypertension. Therefore, RH seems to be a more important prognostic factor than BP control [2]. This is most likely because we are treating a population with a higher risk. However, we cannot discount the significance of controlling BP, since it reduces the absolute CV risk and delays or even prevents the development of organ lesions, such as CKD, which further increase the CV risk in such patients [37]. Contrary to previous research [31,33], a greater number of cerebrovascular events were observed in the present study than acute myocardial infarction. This may be explained by a higher annual incidence of cerebrovascular events in Portugal, compared to the incidence of coronary events [38]. Our findings demonstrate that the 24 h SBP, nighttime SBP, and PP values present the most important prognostic value. In fact, if we compare the SBP and DBP values in the group with events and without events, we find that the difference is verified in the SBP values and not the DBP values. Therefore, we may assume that the discriminative value of the PP will come at the expense of the variation in systolic values. BP changes with age [39]. SBP increases after 40 years of age, and DBP declines after 50 years of age, thus increasing PP [40]. The average age of patients with CV events in this study was 63.2 ± 10.6 years, so it is not surprising that the predictive value of BP is primarily based on SBP values, which will also be reflected in PP. Other studies analyzed the predictive value of ABPM in relation to future CV events in patients with RH [25,41,42,43,44,45,46,47]. Predictive value was reported for SBP values [25,34,42], nighttime SBP [25,41], and daytime SBP [24,41]. Abnormal SBP dipping patterns was also linked to a worst prognosis [44], while no correlation was found for morning surge [48]. One study also found predictive value in DBP values [17]. Finally, the analysis of PP was associated with potential CV events [43]^.^ Those who found significance in SBP values [25,41,42] have a mean age similar to our study, while those who found significance in DBP values [17] were 10 years younger, with a median age of 50 years. With age, the change in BP and PP may be associated with the age-related disruption of the orderly arrangement of elastic lamella in the aorta and central elastic arteries, which stiffens the aorta [39]. In our study, the Kaplan–Meier survival curve clearly demonstrated that patients with a PP >60 mmHg, had worse event survival. PP determined by ABPM data is a well-defined risk factor for CV events [28,30,49]. The European Society of Cardiology/European Society of Hypertension 2018 guidelines considered PP > 60 mmHg as an important marker of worst CV outcome for patients over 60 years [14,40]. According to pathophysiology, PP may be responsible for muscular overload in vessels [50], resulting in stiffer arteries [40] due to an increase in collagen content and a decrease in elastin fibers. This activates the renin-angiotensin system, increasing the inflammation pathway, promoting organ damage, and increasing vessel stiffness, thus becoming a vicious cycle [40]. In 2016, Salles and Cardoso [27] summarized all the longitudinal studies in patients with RH that analyzed the prognostic value of ABPM for future CV events. The authors concluded that nighttime BP and the nondipping pattern were the most significant [27]. In 2021, an analysis of a cohort of 1276 patients with RH over eight three-year periods found that ABPM was the best prognostic marker for morbidity and CV mortality [51]. In both the stated studies, PP was not examined. Lempiäinen et al. [52] studied 1045 arterial hypertension patients undergoing ABPM (56.7% with a CV event). According to their findings, middle-aged people’s high nighttime PP was the most accurate ambulatory BP indicator of CV and all-cause mortality [52]. The same was noted in studies on populations with essential hypertension [28,29]. Our results support the predictive value of PP and the cut off PP > 60 mmHg as a useful tool for the prediction of new CV events in patients with true RH.

## 5. Study Limitations

This retrospective study has some limitations that deserve attention. Firstly, the sample size was relatively small and mainly comprised white/Caucasian patients from a limited geographic region, making these results difficult to generalize for other populations. Secondly, there was a male predominance (approximately 60% of the patients were male). Thirdly, the medications were self-reported at the time of ABPM, which may be inaccurate. In addition, medications can differ between ABPM and follow up. Lastly, follow up was discontinued in 11 patients due to deaths from nonCV events; furthermore, the cause of death was unidentified in six of these patients (absence of cause of death registration in the electronic process). Some of these patients could have died because of a CV event. Taken together, these data can lead to underestimating the outcome of this analysis.

## 6. Conclusions

In the present study, ABPM was essential to obtain a diagnosis of true RH. The SBP and PP are the most important ABPM values for predicting CV events, particularly the 24 h SBP and 24 h PP. For the first time, PP > 60 mmHg was identified as an indispensable tool for determining the prognosis of new CV events in patients with true RH.

## Figures and Tables

**Figure 1 diagnostics-13-01817-f001:**
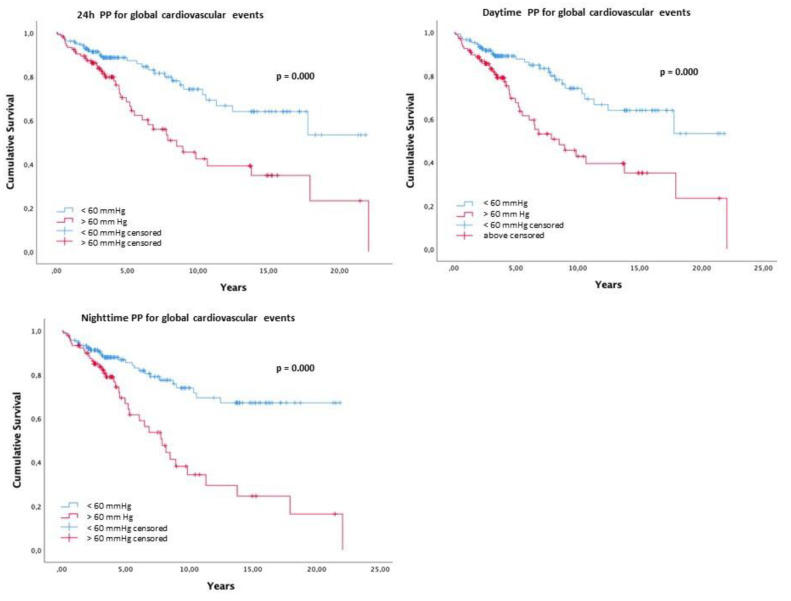
Kaplan–Meier survival curve free of events for the cut off of 24 h, daytime and nighttime PP > 60 mmHg for global cardiovascular events.

**Table 1 diagnostics-13-01817-t001:** Patient characteristics.

	Females (*n* = 100)	Males (*n* = 158)	Total (*n* = 258)
Age (years)	60.4 ± 12.2	60.2 ± 10.5	60.4 ± 11.2
Number of antihypertensive drugs (*n*)	4.0 ± 0.8	4.1 ± 0.8	4.1 ± 0.8
Body Mass Index (kg/m^2^)	29.5 ± 4.7	29.3 ± 4.3	29.4 ± 4.5
Blood glucose (mg/dL)	133.5 ± 54.6	121.3 ± 41.4	126.0 ± 47.2
HbA1c (%)	6.9 ± 1.4	6.7 ± 1.1	6.8 ± 1.2
LDL cholesterol (mg/dL)	106.8 ± 39.3	100.0 ± 34.2	102.5 ± 36.2
Serum Creatinine (mg/dL)	1.3 ± 0.7	1.2 ± 0.4	1.3 ± 1.0
Dyslipidemia, *n* (%)	69 (69%)	123 (77.8%)	192 (74.2%)
Obesity, *n* (%)	44 (44%)	73 (46.2%)	117 (45.2%)
DM, *n* (%)	36 (36%)	81 (51.3%)	117 (45.2%)
Previous CV event, *n* (%)	26 (26%)	63 (39.9%) *	89 (34.4%)
CKD, *n* (%)	36 (36%)	45 (28.5%)	81 (31.3%)
24 h Systolic BP (mmHg)	135.3 ± 18.4	133.9 ± 15.4	134.5 ± 16.6
24 h Diastolic BP (mmHg)	78.2 ± 12.7	76.8 ± 10.1	77.3 ± 11.2
24 h PP (mmHg)	57.2 ± 14.1	57.3 ± 13.5	57.3 ± 13.7
Daytime Systolic BP (mmHg)	140.4 ± 18.7	138.4 ± 15.7	139.2 ± 16.9
Daytime Diastolic BP (mmHg)	82.5 ± 13.4	80.6 ± 10.9	81.3 ± 12.0
Daytime PP (mmHg)	58.1 ± 14.5	58.0 ± 14.1	58.1 ± 14.2
Nighttime Systolic BP (mmHg)	126.7 ± 20.3	124.6 ± 16.5	125.4 ± 18.1
Nighttime Diastolic BP (mmHg)	70.7 ± 13.1	69.5 ± 10.4	69.9 ± 11.5
Nighttime PP (mmHg)	56.1 ± 14.4	55.4 ± 13.0	55.6 ± 13.6
Nocturnal Systolic dipping (mmHg)	10.0 ± 8.4	9.6 ± 9.9	10.6 ± 15.9

BP: blood pressure; CKD: chronic kidney disease; CV: cardiovascular; DM: Diabetes Mellitus; HbA1c: glycated hemoglobin; LDL: low-density lipoprotein; PP: pulse pressure. Blood pressure values were obtained from 24 h ambulatory blood pressure monitoring. * *p* < 0.05.

**Table 2 diagnostics-13-01817-t002:** Cardiovascular event characterization.

Events	Nonfatal Events	Fatal Events	Total
Cerebrovacular events	23	2	25
Coronary events	17	3	20
Other CV events	21	1	22
Sudden Death		1	1
Total	61	7	68

CV: Cardiovascular.

**Table 3 diagnostics-13-01817-t003:** Comparison between the group with events vs. group without event.

	With Event (*n* = 68)	Without Event (*n* = 190)	*p* Value
Age (years)	63.2 ± 10.6	59.3 ± 11.2	<0.05 *
Male, *n* (%)	47 (67.1)	111 (59.0)	0.25
BMI	30 ± 4.4	29 ± 4.5	0.149
Blood glucose (mg/dL)	127.6 ± 43.6	125.5 ± 48.4	0.78
Previous CV event, *n* (%)	37 (52.9)	52 (27.7)	<0.05 *
Obesity, *n* (%)	35 (50.0)	82 (43.6)	0.40
Tabacco, *n* (%)	12 (27.3)	32 (16.8)	0.57
DM, *n* (%)	38 (54.3)	79 (42.0)	0.09
Dyslipidemia, *n* (%)	53 (75.7)	139 (73.9)	0.87
LDL-cholesterol (mg/dL)	96.4 ± 43.6	104.4 ± 33.4	0.18
Serum Creatinine (mg/dL)	1.5 ± 1.8	1.3 ± 0.6	0.09
GFR (mL/min/m^2^)	63.1 ± 32.9	74.7 ± 33.2	<0.05 *
CKD, *n* (%)	32 (52.5)	49 (31.2)	<0.05 *
Number of drugs (*n*)	4.0 ± 0.88	4.1 ± 0.73	0.54
Left Atrium size (mm)	42.3 ± 5.3	40.3 ± 4.8	0.06
LVM gr/m^2^ (*n*)	246 (30)	232 (91)	0.34
Ejection fraction % (*n*)	60.0 (32)	60.9 (97)	0.71
24 h SBP (mmHg)	138.9 ± 16.4	132.8 ± 16.4	<0.05 *
24 h DBP (mmHg)	77.5 ± 10.9	76.8 ± 11.9	0.62
24 h PP (mmHg)	62.3 ± 14.7	55.4 ± 12.9	<0.05 *
24 h P > 60 mmHg, *n* (%)	40 (57.1)	65 (34.6)	<0.05 *
Daytime SBP (mmHg)	143.3 ± 16.1	137.7 ± 17.0	<0.05 *
Daytime DBP (mmHg)	80.6 ± 12.5	81.6 ± 11.8	0.56
Daytime PP (mmHg)	63.0 ± 14.9	56.2 ± 13.6	<0.05 *
Daytime PP > 60 mmHg, *n* (%)	41 (58.6)	68 (36.2)	<0.05 *
Nighttime SBP (mmHg)	130.1 ± 17.6	123.7 ± 17.9	<0.05 *
Nighttime DBP (mmHg)	69.5 ± 12.5	70.1 ± 11.2	0.75
Nighttime PP (mmHg)	60.8 ± 13.8	53.8 ± 13.0	<0.05 *
Nighttime PP > 60 mmHg, *n* (%)	37 (54.4)	54 (29.2)	<0.05 *
SBP nocturnal dipping (mmHg)	8.1 ± 8.9	11.6 ± 17.8	0.14

BMI: Body Mass Index; BP: blood pressure; CKD: chronic kidney disease; CV: cardiovascular; DM: Diabetes Mellitus; DBP: diastolic blood pressure; GFR: glomerular filtration rate; HbA1c: glycated hemoglobin; LDL: low-density lipoprotein; LVM: echocardiographic left ventricular mass; PP: pulse pressure; SBP: systolic blood pressure; n: number of patients; PP > 60 mmHg represents number and % of patients that have PP above 60 mmHg. * Statistically significant *p* < 0.05.

**Table 4 diagnostics-13-01817-t004:** Univariate and multivariate Cox analysis for cardiovascular events, adjusted for confounding variables (age, gender, body mass index, diabetes, and previous cardiovascular events).

Cardiovascular Events	Hazards Ratio (95%CI)
Univariate Cox Analysis	
SBP 24 h	1.37 (1.08–1.72) *
DBP 24 h	0.86 (0.66–1.13)
Daytime SBP	1.29 (0.99–1.68)
Daytime DBP	0.86 (0.66–1.12)
Nighttime SBP	1.45 (1.13–1.88) *
Nighttime DBP	0.92 (0.70–1.21)
24 h PP	1.51 (1.20–1.89) *
Daytime PP	1.44 (1.13–1.82) *
Nighttime PP	1.45 (1.13–1.80) *
PP > 60 mmHg	2.31 (1.42–3.75) *
Daytime PP > 60 mmHg	2.42 (1.49–3.94) *
Nighttime PP > 60 mmHg	2.89 (1.76–4.73) *
Multivariate Cox Analysis	
SBP 24 h	1.44 (1.10–1.88) *
DBP 24 h	1.18 (0.85–1.63)
Daytime SBP	1.31 (0.99–1.75)
Daytime DBP	1.18 (0.86–1.61)
Nighttime SBP	1.35 (1.01–1.80) *
Nighttime DBP	1.12 (0.81–1.55)
24 h PP	1.39 (1.02–1.89) *
Daytime PP	1.22 (0.92–1.63)
Nighttime PP	1.18 (0.90–1.56)
PP > 60 mmHg	1.95 (1.01–3.45) *
Daytime PP > 60 mmHg	2.15 (1.21–3.83) *
Nighttime PP > 60 mmHg	2.07 (1.17–3.67) *

DBP: Diastolic blood pressure; PP: pulse pressure; SBP: systolic blood pressure; SD: standard deviation. * *p* < 0.05.

## Data Availability

Not applicable.

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
