# Peer review of "Systolic Blood Pressure and Pulse Pressure Are Predictors of Future Cardiovascular Events in Patients with True Resistant Hypertension"

_diagnostics, 2023, doi:10.3390/diagnostics13101817_

Round 1
Reviewer 1 Report
Summary
The aim of this study was to evaluate the prognostic relevance of 24-hour systolic blood pressure (SBP) and 24-hour pulse pressure (PP) in resistant hypertension (RH) diagnosed by ambulatory BP monitoring (ABPM). This study included 258 patients (mean age 60 years; 61 % male) who underwent 24-h ABPM from 1999 to 2019 and met the RH criteria. A composite endpoint including cerebrovascular, coronary, and other events was evaluated. The mean follow-up period was 6.0 ± 5.0 years. Seventy events (61 non-fatal and 9 fatal) were recorded. The standard deviation of 24h SBP and of PP were independent predictors of the composite endpoint. The authors concluded that APBM is a fundamental tool for RH diagnosis and for predicting future CV events.
Comments
Introduction: “The 2018 European Society of Cardiology (ESC) hypertension guidelines define resistant hypertension as uncontrolled blood pressure (BP) despite the administration of optimum doses of three first-line classes of antihypertensive drugs (renin-angiotensin system blockers, calcium-channel blockers, and thiazide diuretics) or adequate BP control requiring four or more antihypertensive drugs from different classes [6].” If I am correct, the definition of Resistant Hypertension of 2018 ESC guidelines is “Hypertension is defined as resistant to treatment when the recommended treatment strategy fails to lower office SBP and DBP values to <140 mmHg and/or <90 mmHg, respectively, and the inadequate control of BP is confirmed by ABPM or HBPM in patients whose adherence to therapy has been confirmed. The recommended treatment strategy should include appropriate lifestyle measures and treatment with optimal or best-tolerated doses of three or more drugs, which should include a diuretic, typically an ACE inhibitor or an ARB, and a CCB.”, whereas, the definition of Resistant Hypertension of 2018 ACC/AHA guidelines is “The diagnosis of resistant hypertension is made when a patient takes 3 antihypertensive medications with complementary mechanisms of action (a diuretic should be 1 component) but does not achieve control or when BP control is achieved but requires ≥4 medications.” Please, check your statement.
Globally, the Introduction section could be improved.
The aim of the study should be better reported. Do the authors want to evaluate which blood pressure parameter (systolic blood pressure, diastolic blood pressure or pulse pressure) best predicts prognosis in the context of resistant hypertension?
Abstract, “…from 1999 to 2019…” and Methods, “… between 1999 and 2018…”. Please, check.
Methods: definition of events should be better described by using international definitions and references.
Methods, ABPM : “Recordings with >75% valid data were…” ESC guidelines require > 70%. Please, explain.
Methods, ABPM. “The night-time period was defined as 11:30 pm to 6:30 am, based on the habits of the Portuguese population.” I feel this aspect could be omitted, or you could state that nighttime was defined as the sleeping period.
Methods, ABPM: “Guidelines suggest that in middle-aged and older people, increased PP is associated with an adverse prognosis [6]. In older people, a PP > 60 mmHg is a marker of CV risk [6]. The limits and definitions follow current guidelines and statements [6, 11, 19]. These statements in this section are unclear to me. Please, try to explain better what you mean.
Statistical analysis: survival analyses and univariate and multivariate Cox regression analyses should be better described.
Please check Table 1 and its Title; follow-up should not be reported in this Table; please, show only males or females; please, show parameters aligned to the left of the Table. Please, use . and not , (for example 4.1…..).
The specific number and type of events should be better described.
In the abstract you report there were 70 events, but in Table 2 you report 68 events. Please explain.
Please, check Table 2 format (see comments for Table 1). Marks for statistical significance are not necessary where you report < 0.05; you report left atrium size but you do not report left ventricular hypertrophy and left ventricular ejection fraction; lines between parameters should not be reported.
Table 2: 24h PP<>60mmHg, n (%) and Night-time PP<>60mmHg, n (%). Please, explain.
The Results section reporting survival analyses and Cox Regression analyses should be rewritten. The reason to choose a cutoff of 60 mmHg in your study should be explained. The figures reporting survival curves are of poor quality. In a Table, you should report the results of univariate analyses, including various factors and BP parameters, that is, daytime, nighttime and 24-hour systolic BP, diastolic BP and PP. In multivariate analyses you use age, sex, BMI, DM, previous CV events, glycaemia, creatinine, and the number of antihypertensive drugs. It is unclear to me why you include both diabetes and glycaemia, and why you do not include LDL cholesterol and smoking habit. Please, explain the reason for including the abovementioned variables in the multivariate Cox regression analyses: did you use a group of known risk factors or were these factors associated with risk in univariate analyses? It is not clear how many groups the variable “number of antihypertensive drugs” includes (3, 4, 5 drugs or other?). You had 70 events and you included 8-10 covariates in multivariate analyses. In such a context, the event-per-variable ratio is appropriate for the composite endpoint, but not for separate endpoints (too few events). Thus, you should not analyze separate endpoints (the event-per variable ratio is too low and results are not reliable). Moreover, you should confirm the independent prognostic relevance of PP above the selected cutoff value in multivariate Cox regression analyses and you should not report survival curves only.
I feel that mean arterial pressure should be included in the multivariate analysis.
In the Discussion section, the main findings of this manuscript should be better reported. In the context of RH diagnosed by ABPM, what is the blood pressure parameter that best predicts the risk and possibly what are the therapeutic implications?
The independent prognostic value of the various BP parameters, assessed as continuous or categorical variables, must be confirmed in multivariate Cox analyses (in appropriate models) and cannot be based on Kaplan-Meier curves alone.
In the literature, there are studies reporting the prognostic value of RH diagnosed by ABPM that were not cited.
Pulse pressure is the difference between systolic and diastolic blood pressure, and prognosis may be related to high systolic blood pressure, low diastolic blood pressure, or both. If you want to demonstrate how important or if it is only important the systolic blood pressure you should make a model where you adjust for the diastolic blood pressure and see how the results change. If you want to demonstrate how important or if it is only important diastolic blood pressure you should make a model where you adjust for systolic blood pressure and see how the result changes. If you say that it is only systolic blood pressure that matters, then it is useless to evaluate pulse pressure.
You report a PP cutoff > 60 mmHg. Patients with the same pulse pressure can have quite different systolic and diastolic blood pressure. Do you think their risk profile is the same?
The age of this population was 60.4±11.2 (the range should be approximately 40-80 years) and the follow-up was 6.0±5.0 years. Did diastolic blood pressure values ​​matter in patients younger than 60 years?
It is not clear to me whether patients lost during follow-up were excluded from any type of analysis.
The Discussion section needs substantial changes.
Globally, the manuscript should be improved.
English language should be improved.
Please, check Ref. 1 and Ref. 22 (they are the same).
Please, check reference format.
Author Response
Many thanks for all the sugestions.
Our answers were attached

Reviewer 2 Report
The current study aimed to determine the prognostic value of out-of-office blood pressure monitoring (ABPM) variables in CV prognosis of individuals with resistant hypertension.
General comments:
The use of abbreviations should be introduced appropriately. Abbreviation for resistant hypertension should be introduced first in the introduction before using it.
The manuscript needs to be language edited.
Specific comments:
Methods of the study are not written in much detail and more information is needed:
It is not clear how participants were recruited for the study and how many were included in this analysis.
Details regarding how demographic data, body composition, use of medication and biochemical analyses needs to be included on the manuscript.
Was ethical approval obtained for this study? Did participants give consent for their medical files to be reviewed and data used for this study?
Under the ABPM section on methods, authors referred to the calculation of “nocturnal SBP drop” which I assume is nocturnal SBP dipping, however, on Table 1, this variable is not shown except for “Nocturnal Systolic dipping” which is not clear if it is the SBP drop that is referred to in the methods. Are these variables the same?
Statistics regarding various CV events that are mentioned on the results section, can be presented in a figure or table as this information forms part of the aim of the study.
The resolution of Figure 1, 2, 3 and 4 is low and the reviewer was not able to see the presented information properly to make informed comments on results and discussion of these Figures.
Author Response
Many thanks for the suggestion.
We send in attachement the changes done

Round 2
Reviewer 1 Report
The authors answered various questions and the manuscript has improved . However, I feel some aspects could be further improved.
Abstract: “Seventy cardiovascular events (63 non-fatal and 5 fatal) were recorded”. I think 63 + 5 adds up to 68 and not 70 events. Please check your data.
Abstract: Adjusted HR and 95% CI for cardiovascular events for each SD of 24-hour systolic BP and 24-hour PP should be reported, as well as adjusted HR and 95% CI for cardiovascular events for PP > 60 mmHg as a categorical variable.
Introduction: If you cite only Ref. 7, I feel that this statement should be deleted “….or adequate BP control requiring four or more antihypertensive drugs from different classes”.
Introduction: References from 17 to 27 are not in sequence. Please, check.
Introduction, last sentence: “…which for some authors is approximately 41% more accurate than office blood pressure 7” should be deleted.
Methods: “…or had a controlled BP with four or more antihypertensive agents [6].” If you include these patients according to 2018 ACC/AHHA guidelines, you should include this reference in the Introduction section. Please, clarify if you want to use the definition of resistant hypertension according to 2018 ESC/ESH guidelines or to 2018 ACC/AHA guidelines or both.
Methods, Events: definition should be better described. Was the follow-up similar for patients who did not experience cardiovascular events?
Methods, ABPM : “…with at least 20 measurements during the daytime and during the night.” I feel this sentence should be replaced by the following sentence “… with at least 20 valid readings while awake with at least 2 valid readings per hour and at least 7 valid readings while asleep with at least 1 valid reading per hour.”
Statistical analysis: “First, ABPM variables [SBP, DBP and PP) for the entire 24-h period, daytime and night-time, as well as night-to-day ratios] were tested through a stepwise forward method. Those showing significant associations were then entered in the model with the addition of adjustment variables, which include age, sex, BMI, DM, previous CV event. A multivariate logistic regression model was used to control for confounders and to calculate the adjusted odds ratios (ORs) and 95% confidence intervals (CIs) of 1 standard deviation (SD) increment to determine RH-associated comorbid conditions.” This section looks confused. First, BP parameters should be assessed in univariate Cox regression analysis. Why a stepwise forward method? Then, ambulatory BP parameters should be tested in multivariate Cox regression analysis, avoiding to include in the model parameters that are highly correlated (multicollinearity). For example, daytime, nighttime and 24-hour BP and PP should be analyzed separately with other covariates. Then PP and systolic BP or diastolic BP could only be included in the same model if you want to evaluate the dependent or independent value of PP from systolic BP or diastolic BP (if you retain this aspect necessary). Why you used multivariate logistic regression analysis? I feel that univariate and multivariate Cox regression analyses, and calculation of HR and 95% CI, are satisfactory. Survival analysis should precede Cox regression analyses.
Results: “… patients were identified to have RH, according to the ESC/ESH guidelines7”. Please, see above.
Results: “Regarding the 18 deaths during follow-up, 7 were considered CV events (3 ischemic cerebrovascular events, 3 coronary events, 1 acute heart failure, and 1 sudden death) - table 2, and 11 had non-CV causes (8 unknown causes, 1 cancer, and 2 infectious causes). You state there were 7 CV deaths, but you report 3 cerebrovascular, 3 coronary, 1 acute HF and 1 sudden death, that is 8. Please, check the data.
Table 2: I think the columns do not add up correctly.
Table 3: IMVE should be replaced by LVM (left ventricular mass). Was LVM indexed by body surface area or height^2.7?
Results: “Only 1-SD 24h SBP, 1-SD night SBP,…” You should report the significant parameter (SBP…), the SD is the way (unit) you choose to report the increased risk.
Table 3: See above. Moreover the heading should be changed (you report univariate and multivariate analyses) and the variables used in adjusted models should be checked.
Results: “The 1-SD 24h SBP (HR 1.44 (CI 1.10-1.88) was an independent predictor of CV events, even adjusted to 1-SD 24h DBP (HR 1.48(CI 1.09-2.02)) and 1-SD night SBP (HR 1.35(CI 1.01-1.80). Why you adjusted 24-hour SBP for night SBP? (night SBP is a component of 24-hour SBP).
Globally, results could be better reported.
Please, show P values in the Figures.
Discussion: “Previous research indicates, that the correlation between BP control in RH and adverse outcomes is weaker than the correlation between the presence of RH and adverse outcomes 4,7,43. Smith et al. (2014) reported that patients with RH may have a greater BP burden over time, contributing to a higher CV risk, compared to those without RH, indicating that RH is a more important prognostic factor than BP control”. These sentences are unclear to me.
Discussion: “…assuming that the discriminative value of the PP will come at the expense of the variation in s values.” This sentence is unclear to me.
Discussion: “According to the pathophysiology, PP may be responsible for muscular overload in vessels55, making the arteries stiffer due to higher collagen content and degraded anddecreased elastin fibers 47”. This sentence is unclear to me.
Conclusions: “In the present study, ABPM-based RH diagnosis was deemed crucial. The SBP and PP, particularly the increment of 1-SD of 24h SBP and 1- 24h PP and the cut-off of PP> 60 mmHg, are important ABPM variables for predicting new CV events.” Please, see the aforementioned comments. I feel the conclusion could be better reported.
English language should be improved.
Please, check carefully reference format and completeness of the data.
Author Response
Many thanks for the comments. We send in detail the changes we have made

Reviewer 2 Report
The manuscript has improved as compared to the previous draft. Authors addressed all comments from the reviewer.
Author Response
Many thanks for your comments in all this process.
Mesquita Bastos
Round 3
Reviewer 1 Report
The authors answered the majority of the questions and the manuscript has improved. I feel only some minor changes are needed.
Abstract: I feel that HR (95% CI) for 24-hour BP, nighttime BP and 24-hour PP, as continuous variables, is for 1 SD increase (if possible, this aspect should be reported).
Methods: “CV events were subdivided in coronary events (myocardial infarction, coronary angioplasty, myocardial infarction, angina pectoris, coronary bypass or angioplasty), cerebrovascular events (including ischemic and hemorrhagic strokes, and transient ischemic attack), other CV events (peripheral arterial disease and acute heart failure requiring hospitalization) and sudden death. I suggest to rephrase as follows: “CV events were subdivided in coronary events (myocardial infarction, coronary angioplasty, coronary by-pass, angina pectoris), sudden death, acute heart failure requiring hospitalization, cerebrovascular events (ischemic and hemorrhagic strokes, transient ischemic attack) and peripheral arterial disease.”
Methods: “When the cause of death was not specified, it was recorded as an undetermined, and consequently, non-fatal.” This sentence is unclear to me. I suggest to omit it.
Methods: “Those showing significant associations were then entered in the model, adjusted for the confounding variables (age, sex, body mass index, DM, previous CV event) for the multivariate Cox analysis.” I suggest to rephrase as follows: “Those showing significant associations were then entered in a model including confounding variables (age, sex, body mass index, DM, previous CV event) for the multivariate Cox analysis.”
Methods: “For ABPM variables, a 1 standard deviation (SD) increment was used to determine RH-associated comorbid conditions.” I suggest to rephrase as follows: “For ABPM variables, a 1 standard deviation (SD) increment was used to report HR (95% confidence interval).
Results, Table 3: “LVM gr (n) 246(30) 232(91) 0.34” It is unclear to me if you report LVM in g (mean +/- SD), LVMI in g/m^2 (mean +/- SD) or LVH n (%), that is LVMI > 95 g/m^2 in women and > 115 g/m^2 in men. Please, clarify.
Results: “Figure 1 presents ….. Furthermore, for all CV events, nighttime PP >60 mmHg was associated with higher predictive value with a worse survival with CV events compared to those with PP <60 mmHg”. I feel this sentence should be deleted. Indeed, you did not compare the predictive value of 24-hour, daytime and nighttime PP here.
Results: “Table 3 Univariate and Multivariate Cox analysis for cardiovascular events, adjusted for confounding variables (age, gender, body mass index, diabetes, and previous cardiovascular events). I feel this is Table 4. “Cardiovascular events” should be deleted from the heading. Please, replace “Daytime SBP 1.29 (0.99-1,68)*” with “Daytime SBP 1.29 (0.99-1.68)*”; is it significant (*)?
Discussion: “According to the pathophysiology, PP may be responsible for muscular overload in vessels [50], making the arteries stiffer due to an increment in collagen content and decreased elastin fibers [40].” Is the opposite possible? That is, are stiffer arteries associated with higher PP?
Author Response
Thank you for your helpful suggestions; they were very pertinent. In the attachment, we provide the revised version of the article.
